# When to wake up? The optimal waking-up strategies for starvation-induced persistence

**Yusuke Himeoka**, **Namiko Mitarai**\*

The Niels Bohr Institute, University of Copenhagen, Copenhagen, Denmark

\* mitarai@nbi.ku.dk

**Data Availability Statement:** All relevant data are within the manuscript and its Supporting information files.

**Funding:** This research was funded by the European Research Council (ERC) under the

## Abstract

Prolonged lag time can be induced by starvation contributing to the antibiotic tolerance of bacteria. We analyze the optimal lag time to survive and grow the iterative and stochastic application of antibiotics. A simple model shows that the optimal lag time can exhibit a discontinuous transition when the severeness of the antibiotic application, such as the probability to be exposed the antibiotic, the death rate under the exposure, and the duration of the exposure, is increased. This suggests the possibility of reducing tolerant bacteria by controlled usage of antibiotics application. When the bacterial populations are able to have two phenotypes with different lag times, the fraction of the second phenotype that has different lag time shows a continuous transition. We then present a generic framework to investigate the optimal lag time distribution for total population fitness for a given distribution of the antibiotic application duration. The obtained optimal distributions have multiple peaks for a wide range of the antibiotic application duration distributions, including the case where the latter is monotonically decreasing. The analysis supports the advantage in evolving multiple, possibly discrete phenotypes in lag time for bacterial long-term fitness.

## Author summary

Bacteria grow exponentially consuming nutrients, and then starve until the next nutrient is added. During the starvation, the cells enter dormancy and the cells become tolerant not only to starvation but also to other stressors. When nutrients are given to the starved cells, it takes some time before the cells fully "wake-up" and proliferate again. At first sight, it appears that the shorter this lag time the better for the bacteria. However, if the environment may contain another deadly stressor such as antibiotics, it may be better to "over-sleep" until the stressor is gone. Thus, they need to evolve to optimize their waking up strategy in the fluctuating environment. Here we have developed a theory for the optimal strategy for the repeated grow-and-starvation cycles with a fluctuating application of antibiotics. The optimal lag time exhibits a steep transition from immediate wake-up to over-sleep when the severeness of the antibiotics exceeds the threshold. The proposed general framework provides a way to predict the optimal distribution of lag time for various environmental fluctuation, and it may open for possible applications in

European Union's Horizon 2020 research and innovation programme under grant agreement No. [740704] and by the research grant (00028054) from VILLUM FONDEN to NM. The funders had no role in study design, data collection and analysis, decision to publish, or preparation of the manuscript.

**Competing interests:** The authors have declared that no competing interests exist.

administrating drug usage for interventions of pathogenic bacteria as well as cancer therapies where drug tolerance of dormant cells are observed.

## Introduction

When bacterial cells are transferred from a starving environment to a substrate-rich condition, it takes sometime before the cells start to grow exponentially. This lag phase [1] is often considered as the delay during which the cells modify their gene expression pattern and intracellular composition of macromolecules to adapt to the new environment [2–6]. Therefore, the characteristics of the lag phase depend on the growth condition before the starvation, the environment during the starvation, and the new environment for the regrowth. In spite of this complexity, naïvely thinking, reducing the lag time as possible appears to be better for the bacterial species because it maximizes the chances of population increase. However, interestingly, it has been reported that the distribution of the lag time at a single-cell level has much heavier tail than a normal distribution [7–9].

Indeed, having a subpopulation with long lag time can be beneficial under certain circumstances, for instance, when the nutrients are supplied together with antibiotics. This is because antibiotics often target active cellular growth processes and hence dormant, non-growing cells are tolerant of the killing by antibiotics [10–12]. In general, dormant cells tend to be less sensitive to environmental stress, providing a better chance of survival. Therefore, the lag phase can work as a shelter for the cells from lethal stress.

The phenotypic tolerance provided by a dormant subpopulation has been attracting attention as a course of bacterial persistence [12–16]. Operationally persistence can be categorized into two types [14, 16]; type I or triggered persistence, where the dormancy is triggered by external stress such as starvation, and type II or spontaneous persistence, where the cells switch to a dormant state even though the environment allows exponential growth of the population. The spontaneous persistence has been interpreted as a bet-hedging strategy [17–20], where the optimal switching rates between dormancy and growth is proportional to the switching rates of the environments with and without antibiotic. For the triggered persisters, there should also be an optimal lag time distribution under given antibiotics application. Analysis of the optimal lag time should be relevant in understanding bacterial persistence given a recent laboratory experiment with *Escherichia coli* showing that the starvation triggers the dominant fraction of the bacterial persistence [21], as well as their appearance in pathogenic bacteria *Staphylococcus aureus* and its correlation with antibiotic usage [22].

Previously, Friedman *et al.* [23] have conducted an experiment to see whether bacteria can evolve to increase the lag time by an iterated application of the antibiotics at a lethal level. In the experiment, they grew bacteria with fresh media supplemented with an antibiotic. The antibiotic was removed after a fixed time *T* had passed, and the culture was left for one day to let the survived bacteria grow and enter the stationary phase. Then, a part of the one-day culture was transferred to the next culture, supplemented fresh media with the antibiotic. By repeating the procedure, it was found that the mean lag time of the bacteria has evolved to the roughly same length to *T*, which is expected to be optimal for the long-term population growth.

The present work is motivated by this experiment. In their experiment, the antibiotics were applied at every re-inoculation. What will happen if the application of the antibiotic is probabilistic and the duration of the antibiotics application fluctuates? What are the optimal

distribution and the average of the lag time? Is it better for the total growth to split into sub-populations with different lag times?

In the first part of the present paper, we analyze the optimal waking-up strategy under the probabilistic antibiotic application by using a simple population dynamics model. Analytical and numerical calculations show that the evolution to increase the lag time occurs only if the effect of the antibiotics exceeds a certain threshold, at which a discontinuous transition of the optimal lag time from zero to finite happens. We then extend the model so that the population can be split into two subgroups with different average lag time, to show that there is a continuous transition from single-strategy to bet-hedging strategy when changing the antibiotics application probability and time.

The setup is then generalized in the latter part to ask the optimal lag time distribution without specifying the dynamics in the lag phase and the distribution of the antibiotics application time. The optimal lag time distribution is analytically shown to have a finite gap region in which the probability is zero-valued. Also, the optimal lag time distribution for several distributions of the antibiotics application time is concretely computed. Finally, the implication of the calculated optimal lag time distribution to the biologically observed lag time distribution is discussed.

## Results

### Model with a constant rate wake-up

Motivated by ref. [23], we consider the following setup: Bacterial cells are transferred from stationary phase culture to a fresh media where all the cells are in a dormant state. At every re-inoculation, the fresh media is supplemented with the antibiotics with the probability $p$. The antibiotics are removed at time $T$, and after that, the culture will be left to grow long enough time until $t \gg T$ before entering the stationary phase.

We assume that a cell can take two states, namely, the dormant (or lag) state and the growing state. A cell in the dormant state is assumed to be fully tolerant of the antibiotics but cannot grow (this assumption can be relaxed. See S1 Text, Section.1). A cell in the growing state dies at a rate of $\gamma$ if the antibiotics exist in the environment, and proliferates at a certain rate if there is no antibiotic. Here we suppose that the antibiotics are bactericidal, but not bacteriostatic because the application of the bacteriostatic antibiotics just leads to the prohibition of bacterial activities, and then, there is no meaning in discussing the optimal waking-up strategy.

To be concrete, we first analyze a case where the cells in the dormant state transit to the growing state at a constant rate of $1/\lambda$ as in ref. [23]. Here $\lambda$ corresponds to the average lag time of the population, and the lag time distribution follows an exponential function. The transition from a growing state to the dormant state is not considered when there are nutrients in the culture. Hereafter, we set the proliferation rate to unity by taking its inverse as unit of time. Then, the temporal evolution of the population after an inoculation is ruled by a linear ordinary differential equations as follows;

$$\frac{d}{dt}d(t) \;=\; -d(t)/\lambda, \tag{1}$$

$$\frac{d}{dt}g(t) \;=\; \begin{cases} d(t)/\lambda - \gamma g(t) & (t < T) \\ d(t)/\lambda + g(t) & (t > T), \end{cases} \tag{2}$$

where $d$ and $g$ represent the population in the dormant state and growing state, respectively.

The population dynamics of the antibiotic-free case is obtained by setting $T = 0$. We set the initial population to unity without losing generality, i.e., $d(0) = 1$, $g(0) = 0$.

Since $d(t) = e^{-t/\lambda}$, in $t \gg \lambda$ region, $g(t)$, being asymptotically equal to $f(T)e^t$, represents the total population at time $t$. By noting that the population with zero-lag time grows as $\exp(t)$ under the antibiotic-free condition, $f(T) = g(t;T)/\exp(t)$ measures the impact of the antibiotics and the cost of having non-zero lag time as the population loss relative to the exponential growth without the antibiotics and the lag time. Since $f(T)$ is the measure for a single round of the inoculation, with many repetition of this process, the long-term average normalized growth per round $F_I(\lambda; \gamma, p, T)$ is given by averaging $\ln[f(T)]$ over the probability $p$ of the antibiotics application [24–27] as

$$F_I(\lambda; \gamma, p, T) = (1 - p) \ln \left[ \frac{1}{1 + \lambda} \right]$$
$$+ p \left( -T + \ln \left[ \frac{e^{-T/\lambda} - e^{-\gamma T}}{\gamma \lambda - 1} + \frac{e^{-T/\lambda}}{1 + \lambda} \right] \right). \tag{3}$$

Hereafter, we study the optimal lag time $\lambda^*$ which maximizes the fitness $F_I$ for a given environmental parameter set $p$, $T$, and $\gamma$.

It is worth mentioning the meaning of the optimization of $F_I$. The fitness function $F_I$ is defined as the average of the logarithmic growth over the multiple rounds of waking-up experiment. Thus, the optimization of $F_I$ operationally corresponds to picking up the bacterial culture that grows most successfully after multiple rounds of inoculation among a number of parallel cultures where cells in each parallel culture have different values of $\lambda$. We confirm later that the same result is obtained if the cells are selected after every round of growth.

## Optimal lag time

**Linear-wake up model.** First, we address the case where the duration of the antibiotics application has no fluctuation. In Fig 1A, the optimal lag time $\lambda^*$ is plotted as a function of $p$ for various values of $T$, with $\gamma = 1$. The killing rate $\gamma \approx 1$ is biologically reasonable range since it is often found to be the same order of magnitude of the bacterial growth rate [10, 28]. (For the effect of changing the value of $\gamma$, see S1 Fig.) When changing $p$, the optimal lag time $\lambda^*$ shows a discontinuous transition, with a critical $p$ value dependent on $T$ (and $\gamma$). Below each critical $p$ value, the optimal lag time is zero, while above the threshold, it increases with $p$ as $\lambda^* \sim pT$, reaching $\lambda^* \sim T$ when antibiotic is always present ($p = 1$). The fitness $F_I$ is plotted as a function of $1/\lambda$ in Fig 1B, demonstrating the appearance of a local maximum at a positive finite $\lambda$ leading to a discontinuous transition above a critical $p$. We found that the transition takes place by changing one of the parameters among $\gamma$, $p$, and $T$ with keeping the rest constant. Interestingly, in terms of triggering the transition, these three parameters inherently play the same role. Thus, we hereafter use the word "severeness" instead of specifying parameters if changing any of the three parameters leads to qualitatively the same results.

In the $\gamma \to \infty$ limit, it is easy to show that the optimal lag time is given by $1/\lambda^* = (-1 + \sqrt{1 + 4/pT})/2$. For the killing rate $\gamma$ and the probability of the antibiotics application $p$, we give proof for the existence of the critical values at which the transition occurs and an upper bound of the critical $\gamma$ being $1/p - 1$ in S1 Text, Section.7.

Note that, in this setup, the total population is allowed to be infinitesimally small. However, in reality, the population size less than a single cell means extinction. In order to take the discreteness of the number of the cells into account, we also performed an optimization of the fitness with a constraint $(d(t) + g(t)) \geq \delta_{ext}$ where $\delta_{ext}$ represents the population allocated for a

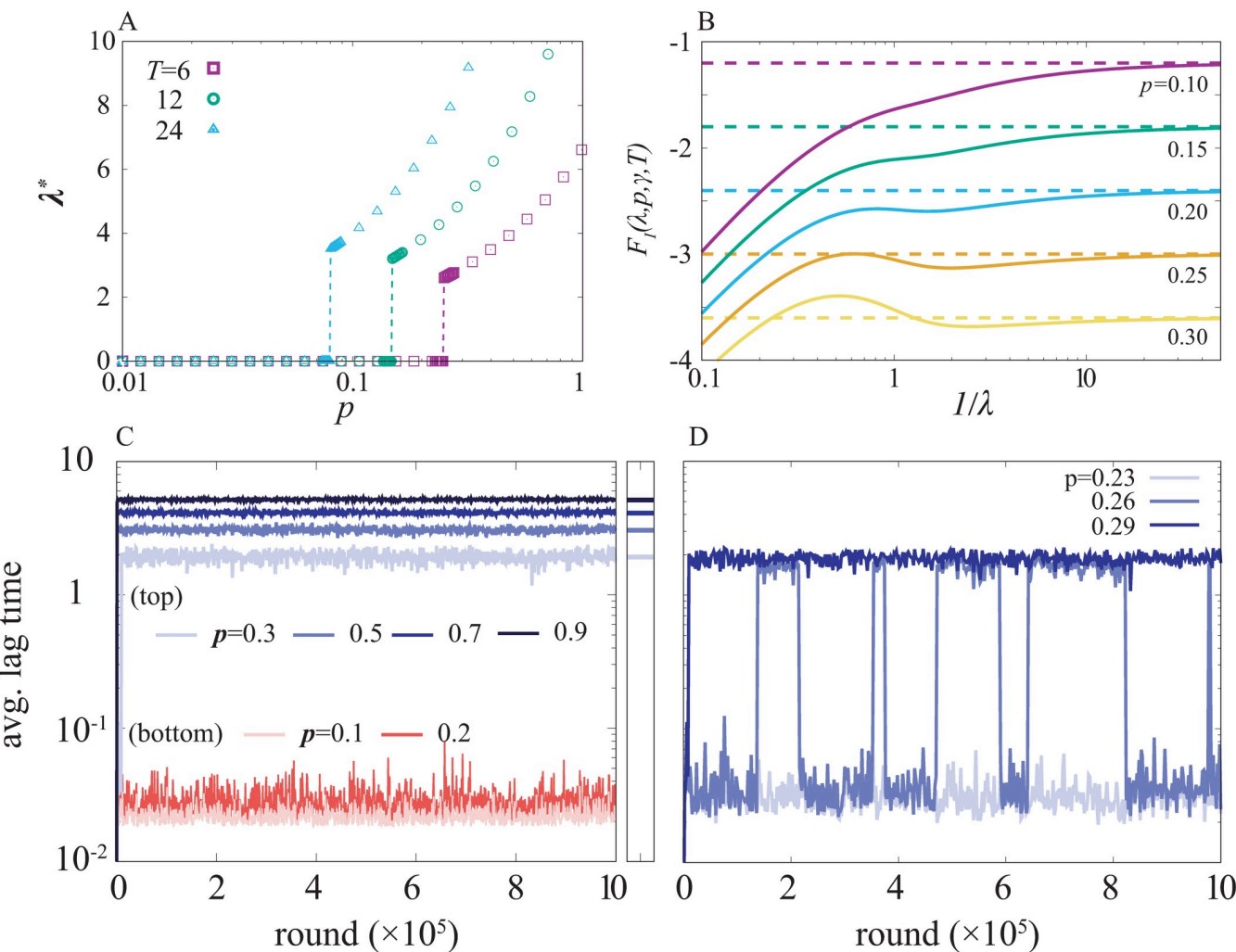

**Fig 1. The optimal lag time value and the evolution of the lag time.** A. The optimal lag time value, $\lambda^*$, is plotted as a function of $p$ for several choices of $T$ value. The optimal lag time shows the discontinuous transition. Below the transition point, $\lambda^*$ is zero. B. The fitness function is plotted as a function of $1/\lambda$ for several values of $p$ close to the critical value. Each dashed line represents $\lim_{1/\lambda \to \infty} F_I(\lambda, \gamma, p, T)$. The local optimal of $F_I$ is formed at $p \approx 0.2$, and it becomes the global optimal at $p \approx 0.25$. C the evolutionary time courses of the averaged lag time in the serial selection model are plotted for several values of $p$. The right panel shows the optimal lag time predicted from the optimization of $F_I$ for corresponding colors while it is not plotted for $p = 0.1$ and $0.2$ because the optimal value is zero. D. The evolution time courses are computed around the critical $p$ value. At $p = 0.26$, the averaged lag time shows a bistable behavior. The figure shares the y axis with C. $\gamma$ is set to unity and $T = 6$ for (b-d). $\Delta\lambda$ (difference of the lag time between the $i$th and $(i - 1)$th population) is set to 0.1 and $N = 200$ for C and D.

single cell in our unit. This modification makes another continuous transition of the optimal lag time from zero to non-zero when $p$ and $T$ are varied, but the model still exhibits the discontinuous transition, too (see S1 Text, Section.2).

**Comparison with sequential selection procedure.**    As we mentioned above, the optimization of $F_I$ corresponds to picking up the bacterial culture that grows most successfully after multiple rounds of inoculation. This is operationally different from an inoculation-dilution cycle performed in ref. [23] where a variety of phenotypes can exist and more fit cells will be selected by the frequency-dependent selection at the stage of the dilution.

In order to ask if the cycle results in different consequences from what we have found by optimizing $F_I$, we performed the inoculation-dilution simulation. The setup of the simulation is the following; we reserve $2N$ variables for the populations of $N$ types of the cells, $d_i$ and

$g_i$ ($0 \leq i < N$), who has the lag time $\lambda_i$ ($\lambda_i = i\Delta\lambda + 10^{-6}$, $10^{-6}$ is added for preventing numerical overflow). Our model cells are incubated *in silico* until $t = \tau$ ($\tau > T$) from the initial state that all the cells are in the dormant state. The cells of the $i$th type wake up at the rate $1/\lambda_i$. The waken-up cells are killed if the antibiotics are applied and $t < T$, otherwise proliferates. When $t$ reaches to $\tau$, the cells are harvested and diluted.

To make the evolution of the lag time possible, we introduced a mutation to the model. When a single cell of the $i$th type divides, the daughter cell can mutate to the $(i − 1)$th or the $(i+ 1)$th type who has a slightly different lag time. Overall, the temporal evolution of the population is ruled by the linear equation (Eqs. (1) and (2)) with a replacement of $\lambda$ by $\lambda_i$ and addition of the mutation term under the growing condition (the detailed equations are provided and extended models are explored in S1 Text, Section.3).

In the dilution process, each type of the cells is diluted proportionally to its fraction in the harvested culture (i.e, $d_i(0)$ for the next round is given as $(d_i(\tau)+ g_i(\tau))/\Sigma_j(d_j(\tau)+ g_j(\tau))$ of the current round, and $g_i(0)$ for the next round is zero.), and thus, it leads to the frequency-dependent selection. Since all the parameters are the same among $i'$s except the lag time, the cells with an adequate lag time is supposed to fit this sequential culture the most and to be selected. We introduced the smallest number of the cell $\delta_{ext}$ also to this simulation. Every time the antibiotics application ends and the dilution is completed, we check the number of the cells of each phenotype and if its value is below the threshold, the value is truncated to zero.

Also, to be consistent with the present model, the antibiotic is applied in probability $p$ and for the duration of $T$. The incubation time $\tau$ is set significantly longer than the studied values of antibiotics application duration $T$ so that there is enough time for exponential growth. Concretely, we studied range of $T$ being from 0.0 to 6.0 and used $\tau = 20.0$.

Fig 1C and 1D show the evolutionary time course of the average lag time $\langle\lambda\rangle$ over the population for several values of $p$. The evolution simulation was initiated from all the population that has the shortest lag time (i.e., $d_0(0) = 1$). As shown in Fig 1C, for the small $p$ values, the population stays at $\langle\lambda\rangle < 10^{-1}$ which corresponds to having the peak at $i = 0$ in the population distribution in $i$ space. In contrast, for $p \geq 0.3$, the average lag time increases over the rounds and settles down at a certain value which is consistent with the value predicted by the optimization of $F_I$. Around the critical $p$ value, the evolutionary dynamics showed a bistable behavior as shown in Fig 1D where the distribution of the lag time in each single round is single-peaked, while it becomes bimodal if we take the avarage over the rounds. The critical $p$ value here is inferred as around 0.26 being reasonably close to the critical $p \approx 0.25$ in Fig 1B.

Also, we have tried another rule of the mutation that any single population can mutate to any other population because some mutations may change the lag time drastically (for the detailed equation, see S1 Text, section 3). As the noise level by the mutation has been increased dramatically, dynamics become much noisier than the previous rule of the mutation, and thus, the averaged lag time ($\lambda_{avg.}$) shows rather continuous change with $p$ (Fig 2A). However, looking at the population-averaged lag time at each round, it stays either the lower side ($\lambda_{avg.\ single\ round} \approx 10^{-2}$) or the higher side ($\lambda_{avg.\ single\ round} \approx 10$) most of the time as shown in Fig 2B. Therefore, the continuous behavior stems from the continuous change of the residual time of the two branches, and the stable lag time changes rather discontinuously with $p$.

**Distributed antibiotic application time.** The discontinuous transition of the optimal lag time observed so far is easy to interpret as a result of competition between the zero lag time being optimal for the no-antibiotic case, while $T$ being optimal when antibiotic is applied. A nontrivial question is then if the transition stays discontinuous when the antibiotic application time $T$ is distributed. Therefore, we generalized the analysis to the case where the antibiotic application time $T$ fluctuates. The fitness function for such cases is defined by replacing the

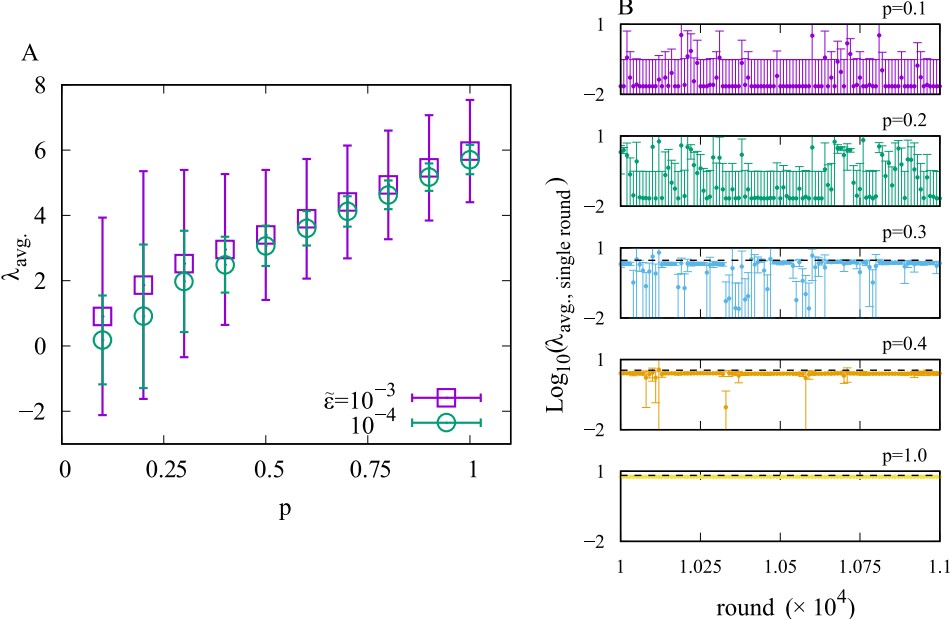

**Fig 2. Random mutation model.** A. Time average of population-averaged lag time of the random mutation model (Eq. 5 in S1 Text) for two choices of the mutation rate ($\tilde{\epsilon}$, see S1 Text section 3). B The population-averaged lag time with one standard deviation of each round (from $10^4$th to $1:1 \times 10^4$th round) for several p values. Data points are plotted every 10 rounds. Error bar indicates the standard deviation. The black dashed line representing the optimal lag time obtained from the optimization of $F_l$ is added for each panel if the optimal lag time is non-zero ($p = 0.3$; 0.4 and 1.0). $N = 200$; $\Delta\lambda = 0.1$, $T = 6.0$ and $\gamma = 1.0$. $\tilde{\epsilon} = 10^{-4}$ for B.

probability of antibiotics application $p$ in Eq (3) by $pq(T)$ where $q(T)$ is the probability distribution function of the duration $T$.

The optimal lag time with a normal distribution with the average $\mu$ and the standard deviation $\sigma$ as $q(T)$ is plotted against the total probability of antibiotics application $p$ in Fig 3. The larger either $\mu$ or $\sigma$ gets, the longer the optimal lag time after the transition becomes. This is simply because an increase of either $\mu$ or $\sigma$ highers the chance of long antibiotics applications to occur.

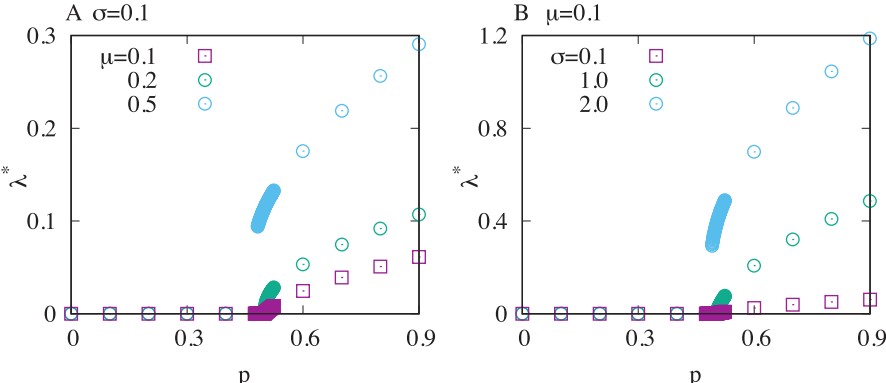

**Fig 3. The optimal lag time for fluctuating T.** The optimal average lag time $\lambda^*$ is plotted as a function the probability of the antibiotics application $p$. A normal distribution with its mean $\mu$ and the standards deviation $\sigma$ is used for the distribution of the antibiotics application time $q(T)$. The transition gets more and more steeper as either A. the average $\mu$ or B. the standard deviation $\sigma$ increases. $\gamma$ is set to unity.

Also, the figure shows that the transition becomes steeper as either $\mu$ or $\sigma$ increases. Indeed, it appears to transit discontinuously for the largest $\mu$ or $\sigma$ case in each panel (cyan data) even though the rest of the data are rather continuous.

Interestingly, in the simplest case where the uniform distribution $q(T) = 1/a$ for $0 \leq T < a$ and $q(T) = 0$ for $T > a$ is chosen, we can analytically show that the transition of the optimal lag time is continuous if $a$ is smaller than $2/\gamma$, but otherwise, it becomes discontinuous (see S1 Text Section.8). This result is consistent with the observation in Gaussian case as increasing the chance of the longer antibiotics application duration triggers the discontinuous transition.

Also, we found that if the lower bound of the support of $q(T)$ is non-zero, the transition of the optimal lag time is always discontinuous regardless of the choice of $q(T)$ (see S1 Text Section.8).

**Multi-step sequential wake-up model.** Finally, we have performed an analysis of an extended version of the model (Eqs (1) and (2)) where the cells sequentially go through multiple ($M$) dormant states as follows:

$$\frac{d}{dt}d_0(t) \;\;=\;\; -d_0(t) \cdot M/\lambda, \tag{4}$$

$$\frac{d}{dt}d_i(t) \;\;=\;\; (d_{i-1} - d_i(t)) \cdot M/\lambda, \;\; (1 \leq i \leq M-1) \tag{5}$$

$$\frac{d}{dt}g(t) \;\;=\;\; \begin{cases} d_{M-1}(t) \cdot M/\lambda - \gamma g(t) & (t < T) \\ d_{M-1}(t) \cdot M/\lambda + g(t) & (t > T), \end{cases} \tag{6}$$

where $M/\lambda$ is the rate of the transition from the $i$th to the $i$+1th state. Note that $M = 1$ corresponds to the original model (Eqs (1) and (2)). The introduction of the multiple steps leads to the Erlang distribution $P(l) = l^{M-1} e^{-lM/\lambda}/((M-1)!(\lambda/M)^M)$ with an average $\lambda$ as the lag time distribution.

Note that the Erlang distribution with $M = 1$ is the exponential distribution that we addressed above. The parameter $\lambda$ is scaled by $M$ in the multi-step model so that the average lag time is unaffected by the number of steps, while the higher moments of the distribution (e.g. variance) changes by the number of steps. Especially, the introduction of more than one step allows distribution to have its peak at non-zero lag time.

In S1 Text, Section.1 and 8, we showed that the discontinuous transition is triggered also in the extended model. The more dormant states the system has, the better the fitness function gets at its optimal $\lambda$ under relatively large $p$ because the distribution is narrower.

Also, the same argument with the single-step model (Eqs (1) and (2)) on the discontinuous transition holds for arbitrary numbers of the intermediate dormant states, i.e., the non-zero lower bound is the sufficient condition for $M$-step sequential models to exhibit the discontinuous transition in the lag time regardless of the choice of $q(T)$ (see S1 Text, Section.8).

## Bet-hedging

So far, we have studied the optimal lag time where all the cells have the same transition rate ($1/\lambda$). However, it is known that even an isogenic bacterial population can split the population into several phenotypes. To see whether the best strategy changes in the multi-phenotype case, we did the simplest extension of the single-step model (Eqs (1) and (2)) to the case where the bacteria is capable of having two subpopulations with different values of average lag time.

We split the total population into two parts, $a$ and $b$, and assume the transition rates from the dormant to growing state being $1/\lambda_a$ and $1/\lambda_b$, respectively. Without loss of generality, we

set $\lambda_a \le \lambda_b$, and denote the fraction of the subpopulation $b$ as $x$. The fitness function is then written down as

$$F_{II}(x, \lambda_a, \lambda_b; \gamma, p, T) = p \ln[(1-x)f_a(T) + xf_b(T)]$$
$$+ (1-p) \ln[(1-x)f_a(0) + xf_b(0)], \qquad (7)$$

where $f_a(T)$ ($f_b(T)$) is $\ln[g(t;T)/\exp(t)]$ with the transition rate $1/\lambda_a$ ($1/\lambda_b$). In this case, the parameters that the bacteria can evolve to optimize are $\lambda_a$, $\lambda_b$, and $x$.

Fig 4 shows the optimal parameter values as a function of the antibiotics application time $T$ for different probability $p$, obtained by the numerical optimization of $\lambda_a$, $\lambda_b$, and $x$. The optimal lag time of the single-strategy case is also shown for comparison.

Interestingly, the probability of the antibiotics application changes the qualitative behavior of the optimal fraction ($x^*$) to an increase of $T$. For a small $p$ value (Fig 4A and 4C), all the cells have zero lag time in the short $T$ region ($x^* = 0$), while as $T$ gets longer, the population start to invest subpopulation to the non-zero lag time phenotype. This strategy corresponds to the bet-hedging strategy which is investigated by Kelly as an extension of the information theory to gambling [24], and later, studied widely for instance, in the population dynamics field [25, 26] including type-II persistence [17, 18] as well as the finance [29] and the relation between the information theory and biological fitness [27]; as the risk of the antibiotic application becomes larger with longer $T$, it pays off to save a small fraction of the population to finite lag time to hedge the risk. On the other hand, if the value of $p$ is relatively large (Fig 4B and 4D), all the population has non-zero lag time ($x^* = 1$) even if $T$ is small, reflecting that the chance of having antibiotics is too high that it does not worth betting subpopulation into zero lag time going for more growth in no-antibiotic condition. However, as $T$ gets larger, the optimal strategy changes to bet some fraction of the population to zero-lag time. This somewhat counterintuitive result is due to the trade-off between the benefit and cost of having a longer lag time. Cells can avoid getting killed by having the lag time being close to the antibiotics application time $T$. However, having a long lag time means waiving the opportunity to grow even when the fresh media is fortunately antibiotics-free. While the loss of the opportunity is negligible for small $T$, as $T$ gets larger, the loss becomes sizable and it becomes better for the population to bet some fraction for the chance of the media to be antibiotics-free.

In the analysis, we also found some locally optimal strategies, which are shown in Fig 4A and 4B as $x_L^*$. For both $p = 0.2$ and $p = 0.8$, the optimal strategy for short $T$ is having a single phenotype ($x^* = 0$ for A and $x^* = 1$ for B). For $p = 0.2$ case, the single phenotype strategy is locally stable as long as the zero-lag time is the optimal lag time for that strategy ($x_L^* = 0$), while after the optimal lag time for single phenotype case $\lambda^*$ becomes non-zero, the single-strategy is no longer optimal, even locally. On the other hand, for $p = 0.8$ case, the single phenotype strategy remains locally optimal after the transition in a whole range shown in the figure.

In Fig 5, the globally optimal fraction $x^*$ of the subpopulation with finite lag time is shown as a heat-map as functions of $p$ and $T$. There are three phases, namely, (i) all the cells waking up immediately ($x^* = 0$), (ii) all the cells have the finite lag time ($x^* = 1$), and (iii) the bet-hedging phase ($0 < x^* < 1$). The $x^* = 0$ phase and the $x^* = 1$ phase are placed in the region with small $p$ and $T$ and the region with large $p$ and small $T$, respectively. For the shown case of $\gamma = 1$, behavior of $x^*$ to the increase of $T$ changes at $p \approx 0.5$. Increasing (decreasing) $\gamma$ shifts the phase boundaries to smaller (larger) $p$ and $T$.

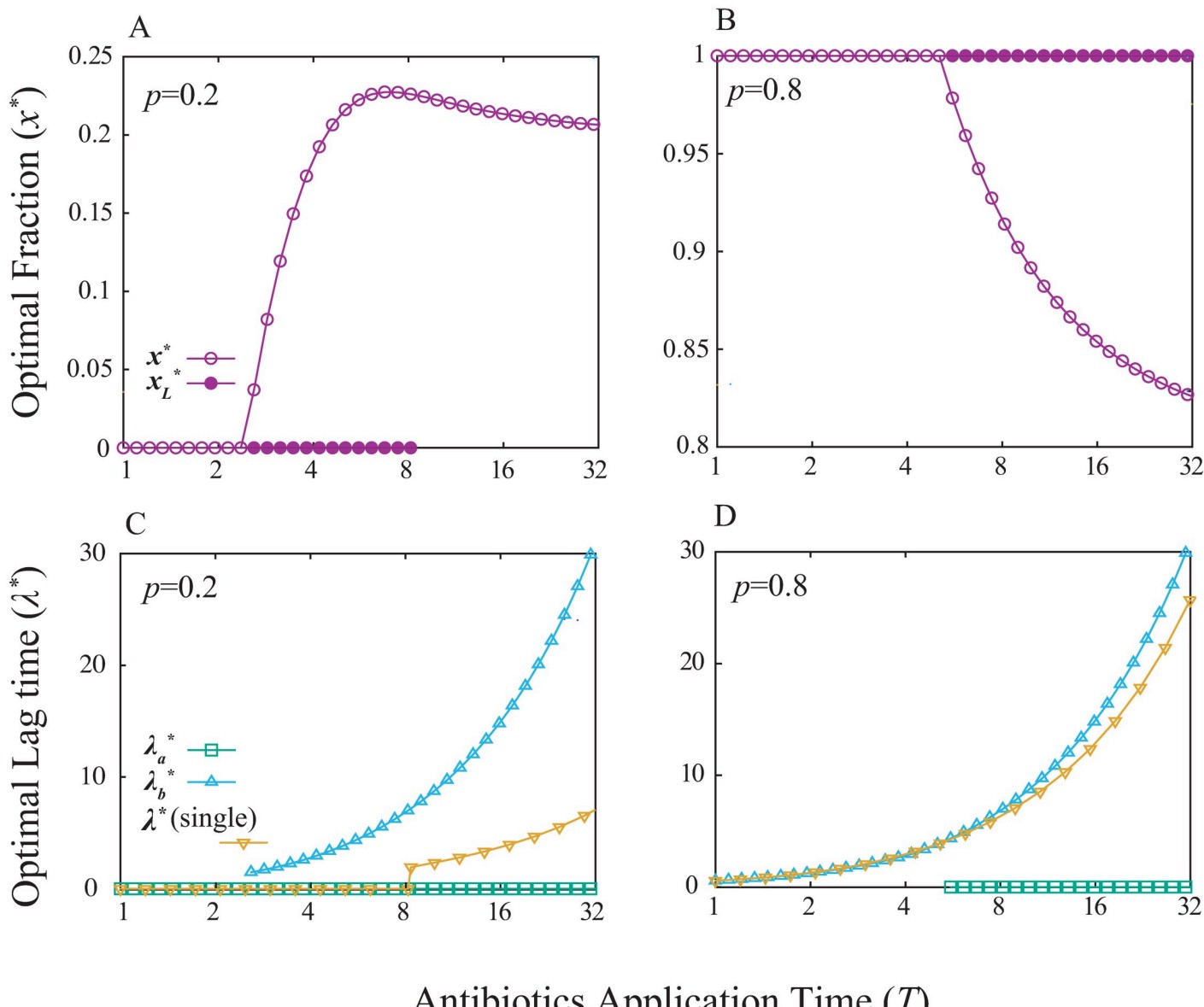

**Fig 4. The optimal strategy for the two phenotypes case.** The optimal fraction of the strategy with non-zero lag time $x^*$ (A and B) and the optimal lag times of both two strategies ($\lambda_a^* = 0$ and $\lambda_b^*$) and the one strategy case ($\lambda^*$) (C and D) are plotted against the antibiotics application time $T$ for different probability of the antibiotics application $p$. In A and B $x^*$ (circles) and $x_L^*$ (filled points) represents the globally and locally optimal fraction, respectively. In C and D, the optimal lag time of the phenotype is plotted only if its globally optimal fraction in the population is non-zero because any value is allowed as the optimal if the fraction is zero. $p = 0.2$ for A and C, and 0.8 for B and D. $\gamma$ is set to unity.

## Optimal lag time distribution

**Generalized set up.** So far, we have studied the optimal average lag time of the bacterial population with a single and two phenotypes by assuming a specific wake up process. The discontinuous transition of the average lag time and the bifurcation of the strategies are shown to be triggered when the effect of the antibiotics application gets severer.

However, the model setup itself strongly restricts the possible strategies. For instance, the present sequential model results only in the Erlang distribution as the lag time distribution

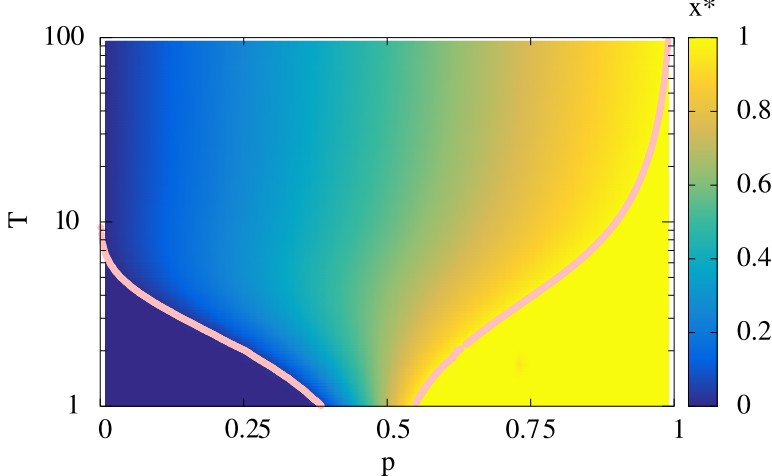

**Fig 5. The phase diagram of the optimal fraction $x^*$.** The solid lines indicate the boundary between $x^* = 0$ and $x^* > 0$ (left region), $x^* = 1$ (right region) and $x^* < 1$. $\gamma$ is set to unity.

where the average and the variance are tightly interconnected and the number of phenotypes equals the number of possible peaks. We, however, do not know the detailed wake-up dynamics or the possible number of phenotypes in real biological systems, hence it is hard to know if the imposed restrictions were reasonable.

Therefore, in this section, we develop a general way to calculate the optimal lag time distribution for a given $T$ distribution, regardless of the internal dynamics of the waking-up, number of phenotypes, and so on. The biological reality will be taken into account afterward to discuss how closer to the optimal distribution the bacterial lag time distribution can approach.

We keep using the same *in silico* experimental setup. Instead of introducing a specific population dynamics model, we consider the lag time distribution $r(l)$. Namely, $r(l)dl$ gives the probability for a cell transit from the dormant states to the growing state at time between $t = l$ and $t = l + dl$. There is no growth or death in the dormant states, while in the growing state, the cells grow at the rate 1 in the no-antibiotic environment, or die at the rate $\gamma$ if the antibiotic is applied. The central assumption is that there is only one growing state and it is an absorbing state: a cell cannot go back to the dormant states once it enters the growing state during one cycle. In other words, $r(l)$ is the first passage time distribution to the growing state in one cycle.

First, we formulate the fitness function. If there is no antibiotic applied, an individual with its lag time $l$ grows after time $t = l$, and thus, the dynamics of the total population without the antibiotics at time $t$, $N_0(t)$ is given as

$$N_0(t)[r] = \int_0^t e^{t-l}r(l)dl + \int_t^\infty r(l)dl, \tag{8}$$

where we assume that the total population is unity at $t = 0$. The first term and the second term represents the cells in the growing and dormant states, respectively.

When the antibiotic is applied for a duration $T$, the cells in the growing state die at the rate $\gamma$ during the duration. The total population $N_T(t)$ at time $t > T$ is then given by

$$\begin{aligned} N_T(t)[r] &= e^t \left( e^{-T} \int_0^T r(s)e^{-\gamma(T-s)}ds + \int_T^t e^{-l}r(l)dl \right) \\ &+ \int_t^\infty r(l)dl. \end{aligned} \tag{9}$$

Following the earlier argument, $f(T)[r] = \lim_{t \to \infty} \ln[N_T(t)[r]/e^t]$ represents the relative growth of the population having the lag time distribution $r$ under the antibiotics application with the duration $T$. Therefore, the fitness of the population with repeated cycle of starvation and stochastic antibiotic application is given by averaging $f(T)[r]$ over the probability of the antibiotics application itself ($p$) and the duration ($q(T)$). Therefore, the generic definition of the fitness function is

$$F[r,q](\gamma, p) = (1-p) \ln \left[ \int_0^\infty e^{-l} r(l) dl \right]$$
$$+ \quad p \int_0^\infty q(T) \ln \left[ e^{-(1+\gamma)T} \int_0^T e^{\gamma l} r(l) dl + \int_T^\infty e^{-l} r(l) dl \right] dT. \tag{10}$$

Indeed, it results in Eq (3) with $T = T_0$ when we chose $r(l) = \lambda e^{-l/\lambda}$ and $q(T) = \delta(T - T_0)$. The optimal lag time distribution is obtained by finding $r(l)$ that maximizes the fitness function (10).

**General form of the optimal lag time distribution.**   Interestingly, it is proven (the detail in S1 Text, Section.9) that the optimal lag time distribution always has the delta function at the origin (including the case being multiplied by zero) and zero-valued region right next to it. The form of the optimal lag time distribution highlighting this feature is

$$r^*(l) = \alpha \delta(l) + (1-\alpha) s(l),$$

where $\alpha$ ($0 \le \alpha \le 1$) is optimal fraction of the delta function and the rest given as a continuous function of $\gamma$, $p$, and $q(T)$, and $s(l)$ is a piecewise-continuous function with minsupp($s(l)$) being greater than zero. This statement holds regardless of the choice of the antibiotics application time distribution $q(T)$.

Note that the statement includes the case that $\alpha$ becomes zero under certain choices of $\gamma$, $p$, and $q(T)$. Thus, in other words, the statement means that if $r^*(0)$ is non-zero, it never comes from any continuous function, but only from the delta function because minsupp($s(l)$) $> 0$.

Remembering that the optimal lag time distribution with $p = 0$ (no antibiotic application) is always given by $\alpha = 1$, the existence of the gap between the origin and minsupp($s(l)$) means that another peak of the optimal distribution never stems from the delta peak at the origin, but discontinuously appears as $p$, $\gamma$ or $q(T)$ changes.

On the other hand, $\alpha$ (the ratio of the delta peak and other peak(s)) continuously changes with the impact of the antibiotics application. When $0 < \alpha < 1$, $r^*(l)$ has at least two disconnected peaks, which indicates that having at least two distinguishable phenotypes is the optimal strategy. This corresponds to the bet-hedging strategy, where fraction $\alpha$ bets on the no-antibiotic environment to maximize the duration of growth, while the fraction $1 - \alpha$ is hedging to survive the antibiotic application period better.

**Computing the optimal lag time distribution.**   We now summarize a numerical method to obtain the optimal lag time distribution for a given $q(T)$. Here we compute the optimal lag time distribution for $q(T)$ with its upper bound of the support as $T_{\max}$ (Note that the different choices of $T_{\max}$ leads to the different distribution of the antibiotics application time, and accordingly, different optimal lag time distribution. For the comparison of the optimal distribution for the different $T_{\max}$ values, see S2 Fig.). Then, the upper bound of the support of the optimal lag time is shown also to be $T_{\max}$ (see S1 Text, Section.9). Next, we approximate the fitness $F$ by discretizing $r(l)$ and $q(T)$ by bins with the size $\Delta$. The approximated fitness function

is given as

$$F_d(\mathbf{r}, \mathbf{q}, \gamma, p; N, \Delta) = (1 - p) \ln \left[ \sum_{n=0}^{N-1} e^{-n\Delta} r_n \right]$$

$$+ \; p \sum_{m=0}^{N-1} q_m \ln \left[ e^{-(1+\gamma)m\Delta} \sum_{n=0}^{m-1} e^{\gamma n\Delta} r_n + \sum_{n=m}^{N-1} e^{-n\Delta} r_n \right],$$

where $N = T_{\max}/\Delta$. Here, $\mathbf{r}$ and $\mathbf{q}$ are the vectors of the discretized probability distribution functions with the $n$-th elements being $r_n = \int_{n\Delta}^{(n+1)\Delta} r(l) dl / \int_0^{T_{\max}} r(l) dl$ and $q_n = \int_{n\Delta}^{(n+1)\Delta} q(T) dT / \int_0^{T_{\max}} q(T) dT$, respectively. Note that $\Delta \to 0$ limit leads to $F_d \to F$. The detailed description about the discritization is provided in S1 Text Section.9.

The partial derivatives $\{\partial F_d/\partial r_n\}_{n=0}^{N-1}$ converges to the functional variation $\delta F/\delta r$ by taking $\Delta \to 0$ limit because $F_d$ converges to $F$ under this limit and $F$ is the $C^\infty$ functional of $r$. Thus, in principle, one can calculate the optimal lag time distribution $r$ by solving $\partial F_d/\partial r_n = 0$ with given value of $\Delta$.

Together with the constraints $\sum_{n=0}^{N-1} r_n = 1$ and $r_n \geq 0$ represented by the Karush–Kuhn–Tucker (KKT) multiplier terms [30, 31], the KKT conditions to determine the optimal distribution $\{r_n\}_{n=0}^{N-1}$ is

$$1 - \mu_n = \sum_{m=0}^{N-1} q'_m \frac{h_n^m}{\langle h^m \rangle}, \quad (\mu_n = 0 \; \text{or} \; r_n = 0) \tag{11}$$

with $r_n \geq 0$ and $\mu_n \geq 0$, where $\mu_n$ is the KKT multiplier for $r_n \geq 0$. The value of the multiplier for the condition $\sum_{n=0}^{N-1} r_n = 1$ is already fixed in the above expression (see S1 Text Section.9). We here introduced a notation $q'_0 = (1 - p)q_0$ and $q'_n = pq_n \; (n \geq 1)$. $h_n^m$ is given by

$$h_n^m = \begin{cases} \exp\left[-m\Delta - \gamma((m - n)\Delta)\right] & (n < m) \\ \exp\left[-n\Delta\right] & (\text{otherwise}), \end{cases} \tag{12}$$

and $\langle h^m \rangle$ represents the average of $\{h_n^m\}_{n=0}^{N-1}$ over $\{r_n\}_{n=0}^{N-1}$. The $n$th bin has a non-zero value only if the $n$th equation of Eq (11) is satisfiable with $\mu_n = 0$, and otherwise, $r_n$ needs to be zero. Thus, the number of non-zero bins is the same with the number of satisfiable equations with $\mu_n = 0$.

Interestingly, the distribution $\{r_n\}_{n=0}^{N-1}$ appears in Eq (11) only in the form of the average of $h'$s. Therefore, the number of free variables equals the number of averages in the equation. For instance, if $q(T)$ is the Dirac's delta function $\delta(T - T_0)$ with $T_0 > 0$ (i.e., fixed $T$ case), only $q'_0$ and $q'_a \; (a \leq T_0/\Delta < a + 1))$ are non-zero. Thus, there are only two free variables; only two bins, say $r_i$ and $r_j$, can be non-zero valued, while others are zero because the number of satisfiable independent linear equations is equal to the number of independent variables (note that the equation is linear by regarding $1/\langle h^m \rangle$'s as the variables.). Thus, the fixed $T$ leads to the sum of two Dirac's delta functions as the optimal lag time distribution.

By numerically solving the KKT conditions (Eq (11)) for several choices of $q(T)$, the optimal lag time distributions can be obtained. Fig 6 shows the obtained optimal lag time distributions for (A) a normal distribution, (B) an exponential distribution, (C) a power-law distribution, and (D) a sum of two normal distributions as $q(T)$ (the protocol for the computation is described in S1 Text Section.9).

The obtained optimal distribution contains a common feature: they consist of the Dirac's delta function-type peak at the origin and the other part mimicking $q(T)$ with steep peaks at its

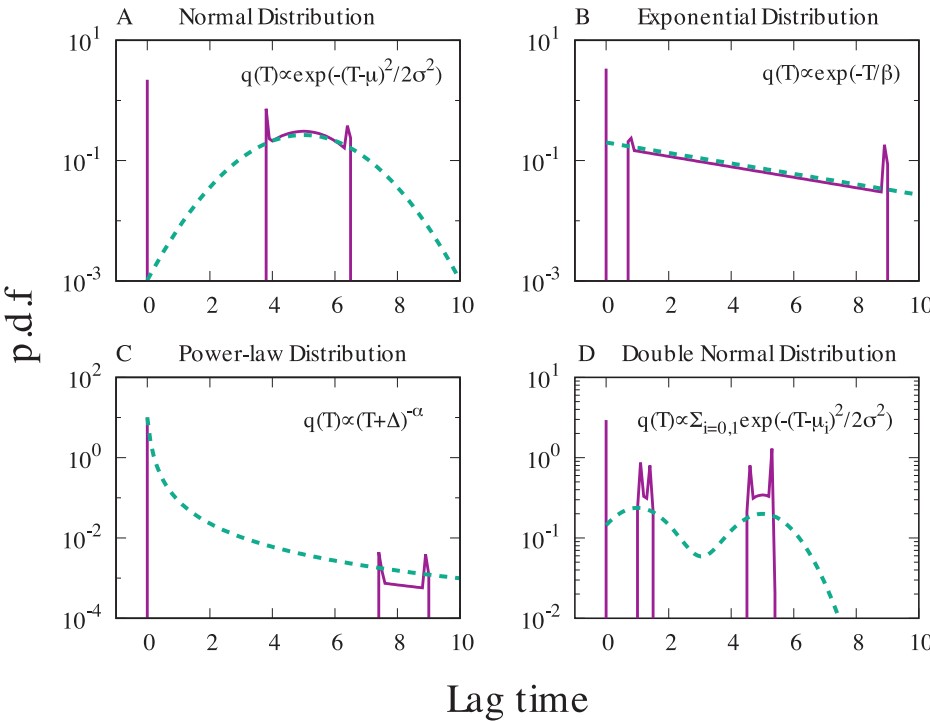

**Fig 6. The optimal lag time distributions.** The optimal lag time distributions (purple solid lines) obtained by solving Eq (11) are plotted. The distribution function $q(T)$ (green dashed lines) is A .a normal distribution with the average 5 and standard deviation 1.5, B .an exponential distribution with the average 5, C .a power-law distribution with saturation, $q(T) \propto (T + \Delta)^{-\alpha}$, where $\alpha$ is 2, and D .a sum of two equal-weighted normal distributions with the average 1 and 5 while the standard deviation is commonly 1.0. Parameters used in the computation are $\Delta = 0.1$, $T_{max} = 10$, $p = 0.8$ and $\gamma = 1.0$.

both sides. Interestingly, the optimal lag time distribution has not only the gap next to the origin as proven but also cut-offs in the upper limit. The "mimicking part" can be further divided into the sum of multiple disconnected functions, and the number of the disconnected regions seems to be the same with the number of peaks of $q(T)$ as long as the peaks are distant to each other.

## Comparison of the optimal lag time distribution and the results using the multi-step sequential wake-up model

Lastly, we compare the obtained optimal lag time distribution with the optimal lag time distribution achieved by the $M$-step sequential model (4)–(6) extended to multiple number ($N_p$) of phenotypes (details are described in S1 Text, Section.5). The lag time distribution obtained in this model is always the summation of $N_p$ Erlang distributions. For a given $M$ and $N_p$, we calculate the fitness value for an Erlang distribution with given average lag time and its fraction in the population for each phenotype by numerical integration to find the optimal lag time values that give the largest fitness value.

Fig 7A and 7B shows the largest fitness value obtained by optimized $M$-state $N_p$ phenotype sequential model as function of $M$ for $N_p = 1$, 2, and 3, with $q(T)$ being a normal distribution and an exponential distribution, respectively. The largest fitness value achieved by the solution of Eq (11) for given $q(T)$ is also shown in the figure. The optimal fitness value is a non-monotonic function of the number of states $M$, while it is increasing function of the number of phenotypes $N_p$ because a population consisting of $N_p$ types can realize any lag time distribution

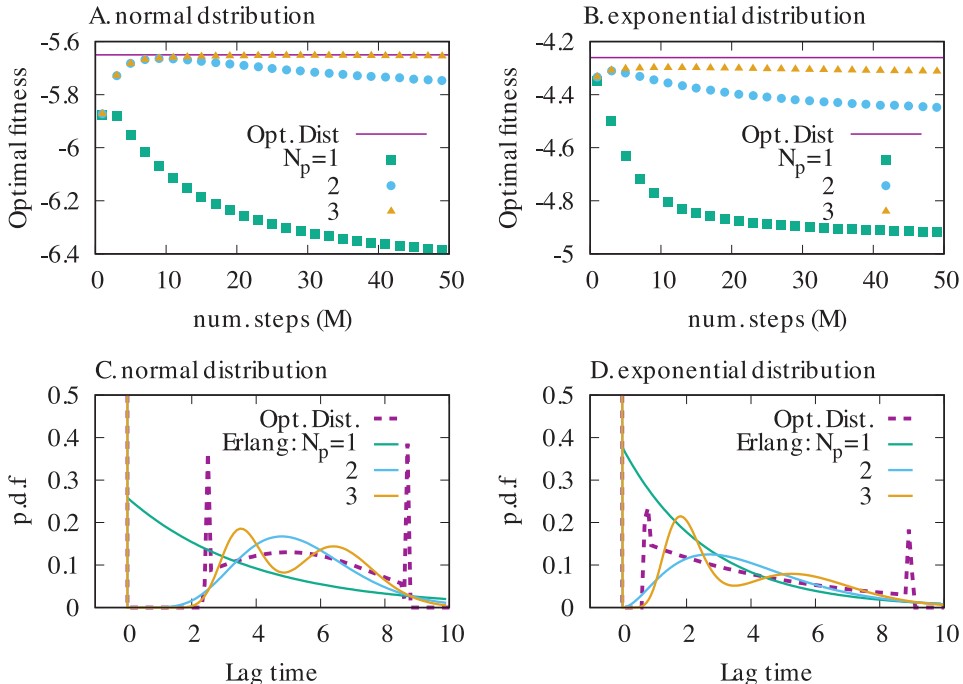

**Fig 7. The optimal Erlang distributions.** The optimal transition rates are computed for given numbers of subpopulations (types) and states for a normal distribution and an exponential distribution as $q(T)$ (B and D). The distribution led by the optimal number of states is plotted for each number of types with the optimal lag time distribution computed from Eq (11) as a reference (A and C). Note that all the distributions with more than one types have the delta function-type peak at the origin. The average and the standard deviation of the normal distribution is 5 and 3, respectively, while the parameter for the exponential distribution is 5.0. The other parameters are given as $T_{max} = 10.0$, $\Delta = 0.1$, $p = 0.8$ and $\gamma = 10$.

achieved by a population with $n_p < N_p$ types. Interestingly, in these examples, the optimal fitness reaches very close to the fitness value achieved by the solution of Eq (11) when there are just two phenotypes ($N_p = 2$) as long as $M$ is chosen appropriately.

The obtained optimal lag time distributions for the number of phenotypes $N_p = 1$, 2, and 3 with $M$ chosen to be the best one to maximize the fitness are depicted in Fig 7C and 7D for the normal distribution and the exponential distribution, respectively. The figure shows that the sequential model with $N_p \geq 2$ is able to successfully capture the essence of the optimal lag time distribution, namely, the delta function at the origin and another peak being distant from the origin.

## Discussion

We have shown that the optimal waking-up strategy changes depending on the severeness of the antibiotic application. When only a single phenotype is allowed and the antibiotic application duration $T$ is a constant, the optimal average lag time exhibits a discontinuous transition from zero lag time to finite lag time as the severeness of the antibiotics is increased (increasing $p$, $T$, or $\gamma$). If the cells can split the population to have two subpopulations with different average lag time, bet-hedging behavior can occur depending on the severeness of antibiotics application, where one sub-population has zero lag time and the other sub-population has a positive finite lag time.

We only presented the analysis of the sequential model for the internal dynamics of the waking-up process, giving the Erlang distribution of lag time for a given phenotype. Another

simple case that analysis is straightforward is the case with $\delta$-function distributed lag time, which is presented in S1 Text, Section.10. There are of course quantitative differences, albeit qualitatively parallel behaviors are obtained in mathematically much clearer form (In the $\delta$-function case, the optimal lag time is either zero or $T$ because waking up just after $T$ can make the whole population avoid death by the antibiotic. This removes the transition in a single phenotype case with changing $T$.).

Further, we have developed a generic expression of the fitness function to study the optimal lag time distribution without assuming a specific model for the waking-up dynamics. We have found that the optimal lag time distribution consists of a weighted sum of a delta function at the origin and some functions with a non-zero lower bound of its support. Importantly, there exists a gap between the delta function at the origin and the rest of the functions. This gap could be the origin of the discontinuous transition of the average lag time in the sequential model with a single phenotype. The continuous change of distributions with only a single peak from the delta function at the origin inevitably creates a non-zero valued area inside the gap region because it is not allowed to make another peak to skip the gap region. Since the gap region is one of the worst parts to invest the population, the only way not to waste their "resource" is to reach the area outside the gap region by the discontinuous (fixed $T$) or steep (distributed $T$) transition.

Of course, the zero-lag time is biologically impossible, hence in reality, what is expected to happen when the zero-lag time is optimal is to evolve to have the shortest possible lag time. The finiteness of the shortest possible lag time affects the location of the critical points as one can infer from Fig 1B, but as long as it is shorter than the finite optimal lag time after the transition, the qualitative nature of the transition stays the same. Also, the zero-lag time appears in the optimal lag time distribution without any assumption on the wake-up dynamics. From the numerical computations, it is expected that the non-delta function part peaks around the average of the antibiotics application time. Therefore, the gap region could be observed if the shortest possible lag time and the averaged antibiotics application time are distinguishably separated.

Assuming that the bacterial cells evolve to reach the optimal lag time, the present analysis implies the nontrivial evolution of tolerant phenotypes after repeated antibiotics application. If it is easier to evolve the population-averaged lag time as single phenotype than having multiple phenotypes with different lag time, then the discontinuous transition predicted in the single-phenotype case implies the following: If the antibiotics are used very often ($p$ higher than the critical probability for the transition), the treatment leads to the prolonged lag time. In contrast, if the antibiotics are used less often (low $p$), the lag time may shorten as a result of selection, even though $p$ is still non-zero. In other words, there is a critical frequency of antibiotic application below which one can avoid the evolution of more tolerant bacteria.

Since the transition is sharp for the single phenotype case, it is expected to be relatively easy to detect the transition. If it is easy to evolve to have multiple phenotypes, the total population-averaged lag time would experience a continuous transition because the transition of the fraction $x^*$ is continuous. If the $p$ and $T$ are kept small enough to stay in $x^* = 0$ phase, the evolution of tolerant phenotype can be avoided. Therefore, detection of the phase boundary can be clinically important.

It is interesting to note that in standard, single-round inoculation experiments without the antibiotics application, a Gaussian distribution in short lag time with an exponential tail in long lag time [7] and a bimodal distribution of lag time [8] have been reported. The observations suggest that relatively clear division into a few phenotypes can happen. That indicates that the bet-hedging strategy of lag time, which shown to be the best solution for the stochastic antibiotics application at the inoculation, can be realized in the bacterial population.

The present analysis has revealed some interesting differences between type-I or triggered persistence studied here and type-II or spontaneous persistence in terms of the optimization problem. The spontaneous persistence has been analyzed as a strategy to cope with the stochastic lethal stress suddenly applied to the environment where cells are already growing exponentially [14, 17, 18]. When multiple phenotypes that have different growth rates for different environments are allowed, the optimal strategy becomes to mimic the fluctuation of the environment change to switch to the best phenotype. This relatively simple outcome is related to the fact that the growth rate difference is amplified exponentially over time. Therefore, the relation between phenotype switching and environmental switching is simple when we assume that the system stays in an environment for more than a few generation time. In the present analysis for the type I or triggered persistence, all the cells start at the dormant state, and the difference in lag time provides a difference in the duration to grow (or die) for a finite time (characterized by typical antibiotics application time $T$), but once the antibiotics are removed there is no difference in the growth condition. This subtle difference is still important as we see that in the experimental evolution of lag time [23], but the trade-off between waking up too early or too late is relatively small over some range of the lag time. This is the reason for the optimal lag time distributions take a fairly non-trivial shape (Fig 6). For interested readers, we made a summary of the correspondence and difference between the analysis of optimal strategy in a fluctuating environment by Kussell and Leibler [18] and the present analysis in S1 Text, Section.6.

In contrast with that the type-II persistence is often described by the analogy of the bet-hedging strategy in gambling [26], the optimal strategy of type-I persistence is possibly analyzed in parallel with the control theory. Finding the optimal lag time distribution is finding the best scheduling when to invest your "budget" (i.e. dormant cells) to wake-up which grows exponentially if the condition is fine to maximize the total growth. This question seems to be compatible with the general framework of the control theory [32, 33] where the system is controlled by time-varying control input to reach the desired state at a given time point. Indeed, the optimal lag time distributions (Fig 6) are reminiscent of the bang-bang control [32–34] where the resources are invested only in a certain period intensively, but no investment for the rest of the time. Furthermore, in the present work, we obtained the discontinuous optimal lag-time distribution, or in other words, the discontinuous support for the distribution function whose integral is constrained to be one. This may be related to the relationship between the $L_1$ constraint and $L_0$ (the support of the functions) constraint in control theory recently studied in the field [35, 36]. An establishment of the relationship between the type-I persistence and the control theory may provide us a much clearer understanding of the type-I and type-II persistence.

As a strategy to cope with the fluctuating environment, the responsive adaptation is also a possibility, where the cells sense the environmental fluctuation and respond to it. The benefits and the costs of sensing the environment are very actively investigated for the cells that are already in a growing state [18, 37, 38]. During the wake-up from the stationary phase, the cells are sensing the nutrients in the environment and responding to it, as it has been shown in the transcriptomic and proteomic analysis of the temporal patterns of the gene expressions during the lag phase [3–6]. If the stress factor in the environment that cells are waking up is not so lethal for the cells in the responding-but-not-fully-growing state, it may be possible for a cell in the lag phase to responsively adapt to the stress. It is an interesting future extension to consider the trade-off of sensing and responsive adaptation in the present setup.

The proposed framework in the present paper provides a way to find the optimal lag time distribution. The actual lag time distribution observed in experiments may be a mixture of the optimality and the restriction imposed on the possible distributions by the physicochemical

constraints, history of the evolution, and other factors. Extraction of the pure optimality by the present framework would help future investigations to unveil those restrictions.

Finally, it should be mentioned that the fundamental understanding of bacterial persistence has an impact on various clinical applications. Recently it has been suggested that the persister formation may enhance the emergence of drug-resistant mutant [39, 40]. Possible mechanisms can be (i) evolution of tolerance supports the rarer resistant mutant to appear [41], (ii) the epistasis between the tolerance and the partial resistance [41], and/or (iii) enhanced mutation rates [42–44] or horizontal gene transfer [45] in the tolerant cells due to stress response programs. Furthermore, accumulating evidence suggests the surprising similarity between the bacterial persistence and antibiotic tolerance of cancer cells [46–54], and in addition, the triggered persistence-like strategy, escaping from the antibiotics efficacy by staying in the inactive phase, was revealed to be the case for cancer cells [54]. It has also been argued in the context of the cancer treatment that drug persistent cells can enhance the appearance of drug-resistant cells [53, 55, 56]. The present analysis provide a starting point of theoretically understand these behaviors.

The present finding shows how type-I persister characteristics can be selected in a relatively simple setup, i.e., the nutrient and antibiotic arrives at the same time and antibiotic disappear before the nutrient is used up. Outside of the laboratory setup, such a scenario may occur when the antibiotic is chemically unstable. The variation of the application duration $T$ may be interpreted as different concentration of the antibiotic applied, which gives different duration for the concentration of the antibiotic to stay above the minimal inhibitory concentration. The proposed framework can be easily extensionded to a more realistic scenario. For example one can consider two kinds of stresses, where one is the starvation, another is the antibiotic application, and let both of the stresses fluctuate. Our present analysis is the limiting case, where the stresses are correlated in a specific manner. In general case, it will be also relevant to consider both the type I (triggered) and type II (sponotaneous) persistence, since if the antibiotic comes in long after the cells are in the nutrient rich condition and nutrient is still available, type II persister is likely more important. We believe our present analysis provide a solid modeling framework that can be extended to such situations for future analysis.

## Methods

The computations have been carried out by Python codes. The KKT conditions (Eq 11) are solved by using `scipy.linalg` and `scipy.optimize.fsolve`. The integrals for obtaining the fitness of Erlang distribution with given parameters in Fig 7A and 7B were computed by using `scipy.integrate.quad`. More details of algorithms are described in S1 Text.

## Supporting information

**S1 Text. Supporting information for "When to wake up? The optimal waking-up strategies for starvation-induced persistence".**
(PDF)

**S1 Fig.** The effect of changing $\gamma$ A. The transition triggered by $\gamma$. B. The transition triggered by $p$ with different $\gamma$ values. $p = 0.2$ for A and $T = 12$ for B.
(EPS)

**S2 Fig. The optimal distribution with different $T_{max}$.** The optimal distribution is computed for A. the exponential and B the power-law distribution, respectively, with two different values of $T_{max}$. Here, we compare the optimal distributions with $T_{max} = 10$ and 12. While the

exponential distribution and the power-law distribution led to the distinct optimal distributions for the different $T_{max}$ values, the optimal distributions for the normal distribution and the sum of the two normal distributions (panel A and D in Fig 6 in the main text) were unchanged. The other parameter values are the same with Fig 6 in the main text.
(EPS)

## Acknowledgments

The authors thank Kenji Kashima for suggesting the references [35, 36].

## Author Contributions

**Conceptualization:** Yusuke Himeoka, Namiko Mitarai.

**Data curation:** Yusuke Himeoka.

**Formal analysis:** Yusuke Himeoka, Namiko Mitarai.

**Funding acquisition:** Namiko Mitarai.

**Investigation:** Yusuke Himeoka, Namiko Mitarai.

**Methodology:** Yusuke Himeoka.

**Project administration:** Yusuke Himeoka.

**Software:** Yusuke Himeoka.

**Supervision:** Namiko Mitarai.

**Validation:** Yusuke Himeoka, Namiko Mitarai.

**Visualization:** Yusuke Himeoka.

**Writing – original draft:** Yusuke Himeoka, Namiko Mitarai.

**Writing – review & editing:** Yusuke Himeoka, Namiko Mitarai.

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
