## [Decision Letter · Decision Letter 0]

21 Sep 2020

Dear Dr. HIMEOKA,

Thank you very much for submitting your manuscript "When to wake up? The optimal waking-up strategies for starvation-induced persistence" for consideration at PLOS Computational Biology.

As with all papers reviewed by the journal, your manuscript was reviewed by members of the editorial board and by several independent reviewers. In light of the reviews (below this email), we would like to invite the resubmission of a significantly-revised version that takes into account the reviewers' comments.

As you shall see, all reviewers recognized the importance of the general topic of the study and appreciated the technical rigor of the presented theoretical and computational analyses. However, the reviewers also raised substantial concerns over the experimental relevance of these analyses and noted insufficient justification of some model assumptions and the corresponding conclusions. These concerns need to be fully addressed if you choose to submit a revision.

We cannot make any decision about publication until we have seen the revised manuscript and your response to the reviewers' comments. Your revised manuscript is also likely to be sent to reviewers for further evaluation.

Sincerely,

Lingchong You

Associate Editor

PLOS Computational Biology

Erik van Nimwegen

Deputy Editor

PLOS Computational Biology

Reviewer's Responses to Questions

**Comments to the Authors:**

Reviewer #1: In this work, Yusuke et al. theoretically analyzed the optimal waking-up strategies for starvation-induced persistence. They first used a simple model with one single phenotype at a constant rate of waking-up. They demonstrated that the optimal lag time shows a discontinuous transition over the severeness of the antibiotic. They then extended the model to have two phenotypes with different lag times and show that a bet-hedging behavior occurs depending on the severeness of antibiotics application. They further presented a generic expression of the fitness function. They found that the optimal lag time distribution consists of a weighted sum of a delta function at the origin and some functions with a non-zero lower bound of its support. Overall, I find that the analysis in this work is very solid, and some of the predictions are very interesting.

Here are some comments.

The prediction of the locally optimal strategies in Fig. 3 is very interesting. Are these locally optimal strategies stable in the experiment? If so, is there any published evidence to support this?

In Fig. 3cd, why are the curves of the λa, λb only shown in some range of the Antibiotics Application Time (T)?

The prediction that the optimal strategy changes to bet some fraction of the population to zero-lab time for larger T is very interesting. Can the authors do some simple simulation to further demonstrate the prediction?

The meaning of the bistable behavior in Fig.1 is not clearly described. Would the distribution of the lag time in one population show bimodal under this case? Is it related to the Bet-hedging case?

Reviewer #2: The manuscript “When to wake up? The optimal waking-up strategies for starvation-induced persistence” presents a computational study of the optimal waking up strategy for bacteria from starvation-induced persistence state when the starvation if lifted and replaced with an environment containing stochastic stress conditions such as antibiotics. The authors initially show that when the stress in the waking up environment is constant, then the optimal lag time is zero if the probability of encountering stress is below a certain threshold, and transition abruptly into a finite value above that threshold, which increases with the probability of encountering the stress. The authors then study a case in which the bacteria have to go through several dormant stages before waking up, and another case where the population is split between two phenotypes, one of which has zero lag-time, while the other has a finite lag-time before waking up. They conclude their study by examining a general case, in which the period of exposure to antibiotics in the waking up environment follows different PDFs and they try to find the optimal corresponding distribution of waking up times that the bacterial population should have to maximize their fitness.

While I find the topic of the manuscript interesting and can be appealing to a large audience, the results presented fall short from convincing me of their importance, and in some cases they are not clear and in others the authors fail to justify their assumptions and conclusions. Below I detail my concerns.

Major concerns:

1) In the section “comparison with sequential selection procedure”, the authors assume that the mutations are continuous and can occur only between closely related phenotypes, which is problematic and not realistic. I wonder what would happen if they considered a random switching case?

2) In figure 2, the authors show that increasing both, the mean exposure-time and its standard deviation, increases the steepness of the transition in optimal lag-time in response to increasing probability of encountering stress. While the result for increasing mean exposure-time might be somewhat clear to me, I do not understand why the increase of standard deviation would have a similar effect. This needs to be explained better.

3) In the section “Multi-step sequential wakeup model”, the authors consider a scenario in which the cells go through several stages before waking up. Since the different stages have exactly the same behavior, and the transition into growth state occurs only from the final stage of the dormant states, I do not see how this model is different from the one dormant state model with different waking up time? The authors need to explain better what is different here.

4) In the section “Bet-hedging” I failed to understand what is xL*?

5) In the section “ General form of the optimal lag time distribution”, the authors assume a specific form of distribution, which seems to split the population into two fractions, one fraction (alpha) that has a lag time zero, and the rest have a distribution of lag times (s(l)). They claim that this is the optimal form and that they prove it in the supplementary info. I could not really see how they prove that. Are they considering in their calculations the possibility that alpha could be zero? I would actually like to see how the fitness change with the value of alpha. It seems to me that this assumption limits the possibilities for s(l), which then appear to follow the distribution of exposure times. This is not surprising, since it actually means that the cells should be more or less responding to their environment. I wonder if for alpha equal to zero, they would get s(l) that completely mimics the distribution of exposure times?

6) Section “Comparison of the optimal lag time distribution and the results using the multi-step sequential wake-up model”. Again, it is not clear what the model is here and how is that different that changing the lag time distribution (see major concern 3).

Minor concerns:

1) Normally, starvation-induced persistence is a response to stress, which can occur in various stressful conditions albeit at different rates. While I think the authors have the right not to consider the possibility of transitioning back to dormant state when the bacteria experience antibiotics, I think that considering this possibility would have made their results more realistic for comparison with experiments.

2) The results presented in the paper are not surprising given the assumptions made by the authors. The authors claim that this study can help in planning antibiotic treatments in order to over come this hurdle of bacteria surviving the treatment through persistence. However, I failed to see how this study contributes to that. In my opinion, it would have been more informative to investigate realistic situations like using real lag-time distributions and find what is the best strategy to kill them.

3) What is minsupp()?

4) Line 221: typo, “bed-hedging” instead of “bet-hedging”.

5) Line 293: typo, “as” should be “is”.

6) Line 320 – 321: typo, “… r and q is the vector of … function… element…” should be “… r and q are the vectors of … functions… elements…”.

Reviewer #3: Lag refers to the time it takes cells to enter growth when transferred from stationary phase to a growth-promoting condition. Most often it is measured in the context of growing a culture to saturation and measuring how long population growth is delayed once cells are reinoculated into fresh media. A consideration of lag-time is important for understanding antibiotic tolerance, as non-growing cells are often more tolerant to antibiotics. In addition, populations of bacteria have been shown to optimize their lag time in response to antibiotic dosing timescales (Fridman, et al 2014)--they resist antibiotics by adjusting their lag to remain in a dormant state until antibiotics have been depleted.

In this manuscript, the authors develop a model (with multiple extensions) that predicts the optimal lag time in response to antibiotic dosing strategies. These models predict the discontinuous transition in average lag time, which has been observed in experiments (Fridman, et al 2014). The model also predicts the experimentally observed phenomena that the lag time evolves to be roughly the same length as the time of antibiotic treatment (T) (Fridman, et al 2014). Other notable predictions of the model are that multiple lag times are optimal for a wide range of antibiotic application distributions, indicating that populations should evolve multiple discrete phenotypes for lag in order to better evolve antibiotic stresses. In addition, they extended their basic model to develop a generic expression for the fitness function to study the optimal lag time distribution without assuming a specific model for the wake-up dynamics. A mapping of their model to the classic Kussell-Leibler model is provided in the supplemental material.

A theoretical understanding of how dormant cells interact with antibiotics is a worthwhile endeavor given the importance for understanding why antibiotic (or chemotherapy, etc) treatments fail. This effort examines a very specific circumstance where fresh nutrients (encouraging growth) and antibiotics are delivered simultaneously, but antibiotics are removed before fresh growth media is. It is hard to imagine how this scenario might play out in a patient. That doesn’t necessarily make the model less valuable, as one needs to make assumptions and see how those play out in the model, but it would be nice to deepen the discussion of what biological phenomena we really expect this model to predict.

Specific Comments:

• What is meant by “severe” when describing the antibiotics treatment? It seems that this could mean lengthening T or changing gamma or p (line 111). These are quite different things in terms of the modeling effort, so I would find it clearer if they were not lumped into one term and if “severe” was not used through the paper but rather “duration, probability, efficacy” or other similarly informative terms.

• Line 100-105: Setting the killing rate to gamma=1—is this reasonable, what would happen as your vary gamma?

• The legend for Figure 1 says “The local peak of the fitness is formed at p~0.2 and the fitness value at the peak exceeds its value at 1/lambda -> infinity”. This does not look true at p~0.2 but does look true as p increases. Please clarify the legend.

• What does “fitters” on line 132 mean? More fit cells?

• The clarification of FI (line 126) should come earlier when FI is first defined. Readers familiar with Kelly bet hedging, etc will already have found this clear, but this will help hold other reader’s understanding.

• I could not find, perhaps I am missing it, the range of lambda used in the simulations for Figure 1C, 1D---what was the jump in lag between populations, and did this matter for the simulation?

• It is not clear what x* and xL* are in Figure 3 (from the figure legend)

• The colormap bar is not labelled on Figure 4---is this x? Would it be easier to label (i), (ii), etc from lines 247-250 on the figure? (Figure 4?)

**Have all data underlying the figures and results presented in the manuscript been provided?**

Reviewer #1: Yes

Reviewer #2: Yes

Reviewer #3: Yes

PLOS authors have the option to publish the peer review history of their article (what does this mean?). If published, this will include your full peer review and any attached files.

Reviewer #1: No

Reviewer #2: No

Reviewer #3: No
---

## [Decision Letter · Decision Letter 1]

21 Dec 2020

Dear Dr. HIMEOKA,

We are pleased to inform you that your manuscript 'When to wake up? The optimal waking-up strategies for starvation-induced persistence' has been provisionally accepted for publication in PLOS Computational Biology.

Before your manuscript can be formally accepted you will need to complete some formatting changes, which you will receive in a follow up email. A member of our team will be in touch with a set of requests. In preparing the final version, we would like you to address the remaining suggestion by reviewer 1.

Best regards,

Lingchong You

Associate Editor

PLOS Computational Biology

Erik van Nimwegen

Deputy Editor

PLOS Computational Biology

Reviewer's Responses to Questions

**Comments to the Authors:**

Reviewer #1: The authors have fully addressed my comments. Thanks.

Reviewer #2: The authors have answered all my previous concerns.

Few minor comments:

1) The random mutation model should be in the main text in my opinion since it reflects the realistic situation and the experimental results that motivated this study better in my opinion.

2) Typos:

a) Line 104, “rounds” should be “round”.

b) Line 121, “leads qualitatively” should be “leads to qualitatively”.

c) Line 197, “rests” should be “rest”.

d) Line 380, “equals to the number” should be “equals the number” or “is equal to the number”.

Reviewer #3: My initial review of this manuscript was overall favorable with a few notable concerns. In this revision the authors have addressed my concerns as follows:

- Clarification of the role of the duration, probability and efficacy parameters (T, gamma, p)

- An additional Supplemental Figure (S1) to clarify the effect of changing gamma

- Removed and edited some confusing text/figure legends and made some additional changes to the text to add clarity and make it more accessible to a wide range of readers. Made additional changes to clarify figures.

- Added needed information regarding the range of lambda explored (and an additional Supplemental Figure-S5)

- Clarified X* and XL* in Figure 3

This manuscript contributes to a theoretical understanding of type-I (triggered) persistence vs type-II (spontaneous) persistence. I agree with the statements of the authors in their response letter regarding the significance of this work. It will provide a theoretical framework for further exploration and understanding of type-I persistence.

**Have all data underlying the figures and results presented in the manuscript been provided?**

Reviewer #1: Yes

Reviewer #2: Yes

Reviewer #3: None

PLOS authors have the option to publish the peer review history of their article (what does this mean?). If published, this will include your full peer review and any attached files.

Reviewer #1: **Yes: **XIAOJUN TIAN

Reviewer #2: No

Reviewer #3: No

---

## [Editor Report · Acceptance letter]

2 Feb 2021

PCOMPBIOL-D-20-01136R1 

When to wake up? The optimal waking-up strategies for starvation-induced persistence

Dear Dr Himeoka,

I am pleased to inform you that your manuscript has been formally accepted for publication in PLOS Computational Biology. Your manuscript is now with our production department and you will be notified of the publication date in due course.

With kind regards,

Alice Ellingham
